# Examination of a nutritional treatment pathway according to pretreatment health status and stress levels of patients undergoing hematopoietic stem cell transplantation

**Takashi Aoyama**[1]*, **Osamu Imataki**[2], **Akifumi Notsu**[3], **Takashi Yurikusa**[4], **Koki Ichimaru**[5], **Masanori Tsuji**[5], **Kanako Yoshitsugu**[5], **Masafumi Fukaya**[5], **Terukazu Enami**[5], **Takashi Ikeda**[5]

1 Dietary Department, Shizuoka Cancer Center, Shizuoka, Japan, 2 Division of Hematology and Stem Cell Transplantation, Kagawa University Hospital, Kagawa, Japan, 3 Clinical Research Center, Shizuoka Cancer Center, Shizuoka, Japan, 4 Division of Dentistry and Oral Surgery, Shizuoka Cancer Center, Shizuoka, Japan, 5 Division of Stem Cell Transplantation, Shizuoka Cancer Center, Shizuoka, Japan

* t.aoyama@scchr.jp

**Data Availability Statement:** All relevant data can be found here: https://figshare.com/articles/dataset/Examination_of_a_nutritional_treatment_

## Abstract

### Introduction

This study aimed to validate hematopoietic stem cell transplantation (HSCT) treatment via a tailored nutritional pathway in myeloablative conditioning (MAC), determine its efficacy in terms of remission, and explore associations between clinical outcomes and nutritional indicators.

### Methods

We included patients who underwent MAC for HSCT at the Shizuoka Cancer Center Stem Cell Transplantation between 2015 and 2019. We evaluated outcomes from the day before treatment initiation (transplant date: day 0) to day 42.

### Results

Among the 40 MAC cases (participant characteristics: 20/40 males, mean age of 52 years, and mean body mass index of 21.9 kg/m$^2$), we found that the percent loss of body weight and loss of skeletal muscle mass were correlated with the basal energy expenditure rate (BEE rate; $r = 0.70$, $p<0.001$ and $r = 0.49$, $p<0.01$, respectively). Based on the receiver operating characteristics curves, the cutoff value for the BEE rate in terms of weight loss was 1.1. Salivary amylase levels did not significantly change during the treatment course. Continuous variables, including oral caloric intake and performance status, showed statistically significant correlations with nutrition-related adverse events during treatment ($r = -0.93$, $p<0.01$ and $r = 0.91$, $p<0.01$, respectively). Skeletal muscle mass before treatment initiation was an independent predictive variable for reduced 2-year survival ($p = 0.04$).

pathway_according_to_pretreatment_health_
status_and_stress_levels_of_patients_
undergoing_hematopoietic_stem_cell_
transplantation1_xlsx/19623993.

**Funding:** This research project was sponsored by
The Foundation for Promotion of Cancer Research
and The Japan Dietetic Association Grants for
Nutritional Guidance-related Research
(2015,2016). This work is supporting by JSPS
KAKENHI Grant Number 22K18237 (2022-2027).
The funders had no role in study design, data
collection and analysis, decision to publish, or
preparation of the manuscript.

**Competing interests:** No.

## Conclusion

Our results support the validity of a safe nutritional pathway with a BEE rate of 1.1 for HSCT patients pretreated with MAC. Specifically, we found that this pathway could prevent weight loss in response to nutrition-related adverse events. Skeletal muscle mass before treatment was identified as an independent risk factor for reduced 2-year survival.

## Introduction

Allogeneic hematopoietic stem cell transplantation (HSCT) [1] for treating hematopoietic tumors aims for remission with enhanced antitumor effects while ensuring immunosuppression. The tumor recurrence rate associated with reduced-intensity conditioning [2] used for HSCT is high compared to that associated with myeloablative conditioning (MAC) [3]. Treatment with reduced regimen-related toxicity is being developed; this treatment replaces cyclophosphamide, a strong emetic, with fludarabine in MAC, and its associated tumor recurrence rate is reportedly no different from that associated with conventional MAC [4–6]. We previously implemented our nutritional pathway and reported the effects of HSCT on medication-related adverse events (in particular, digestive tract adverse events related to nutrition); we also presented more general results regarding this nutritional intervention method [7]. However, because the intervention period was not consistent, we have been unable to suggest an appropriate amount of caloric supply for nutritional treatment [8, 9]. Thus far, there is no consensus in the literature on the correct value of the stress factor (a stress index related to basal energy expenditure [BEE]) in HSCT patients. However, this factor is required to be between 1.3 and 1.5 units, given the increased protein catabolism during fever and steroid administration [10]. To the best of our knowledge, there is no previous study on appropriate nutritional doses considering stress during the course of treatment, nor is there a study on adverse events occurring during MAC in terms of a reduced remission rate [11]. Furthermore, risk factors for adverse outcomes in HSCT treatment in connection with nutritional pathways have not yet been explored. This study aimed to examine the validity of nutritional pathways based on clinical nutritional indicators during a set period of nutritional intervention for MAC during HSCT pretreatment. A secondary endpoint of this study was to explore the associations between clinical nutritional indicators and clinical outcomes. Our interdisciplinary study aims at informing nutritional interventions in patients undergoing HSCT. This important topic is of great interest in the field of oncology.

## Materials and methods

### Participants

Study participants included patients aged 16–70 years with a pretreatment performance status (PS) of 0–1 who underwent HSCT between June 2015 and June 2019 for the first time at the Shizuoka Cancer Center, Division of Stem Cell Transplantation (SCC SCT) [12]. The exclusion criteria were as follows: non-preservation of organ function, pre-treatment of body mass index (BMI) before treatment outside the 18.5–25 kg/m$^2$ range [13], nutritional intervention via a nutritional pathway refusal, transplant-related death, history of reduced-intensity conditioning, history of a second transplantation, organ damage, clinical study cases with duplicate data, engraftment failure, and being unfit for registration in this clinical study as decided by the attending physician due to miscellaneous reasons. Fuludabin/busulfan ≥6.4 mg/kg, cyclophosphamide/total body irradiation ≥8 Gy, and melphalan ≥140 mg/m$^2$ were implemented

for HSCT pretreatment [14]. Bone marrow from unrelated donors, cord blood, and allogeneic peripheral blood stem cells were used as transplant sources.

## Methods for assessing nutritional pathways

The evaluated nutritional pathway (used in the SCC SCT) strictly adhered to food hygiene guidelines from the Japan Society for Hematopoietic Stem Cell Transplantation. These guidelines provide support for patients to return to normal (pretreatment) meals [14, 15]. In the SCC SCT, nutritional support comprises normal (solid) meals and foods. We did not use immunonutrient supplements (glutamine and Arginine) or enhanced nutritional supplements (enteral nutrition) that have not undergone randomized controlled trials with human leukocyte antigens [16].

In the HSCT nutritional pathway evaluated in this study, a registered dietitian went to the patient's bedside every day and inquired about the patient's preferences and chief complaints; these visits were initiated before the treatment and continued until the completion of the parenteral nutrition (PN) with a high-calorie/amino acid preparation. The registered dietitian provided instructions for dietary adjustment and reported the amount of nutrition (i.e., calories, proteins) for PN and oral intake to the attending physician. The diet was then appropriately adjusted based on the patient's chief complaints and symptoms (Fig 1) [17, 18]. This study examined several items (detailed below) between the day before treatment initiation (T1, days 5–8; baseline) and T2 (day 42); the median value for PN administration was selected based on the results of our previous study [8, 9, 17–19].

**Evaluation items.** We evaluated the patient BMI and percent ideal body weight (%IBW) at T1. For the evaluation of percent loss of body weight (%LBW) between T1 and T2 [20] (using InBody S20® Bioelectrical Impedance Analysis [BIA], a high-precision body composition analysis device; InBody Co., Ltd., Seoul, South Korea), we examined skeletal muscle mass

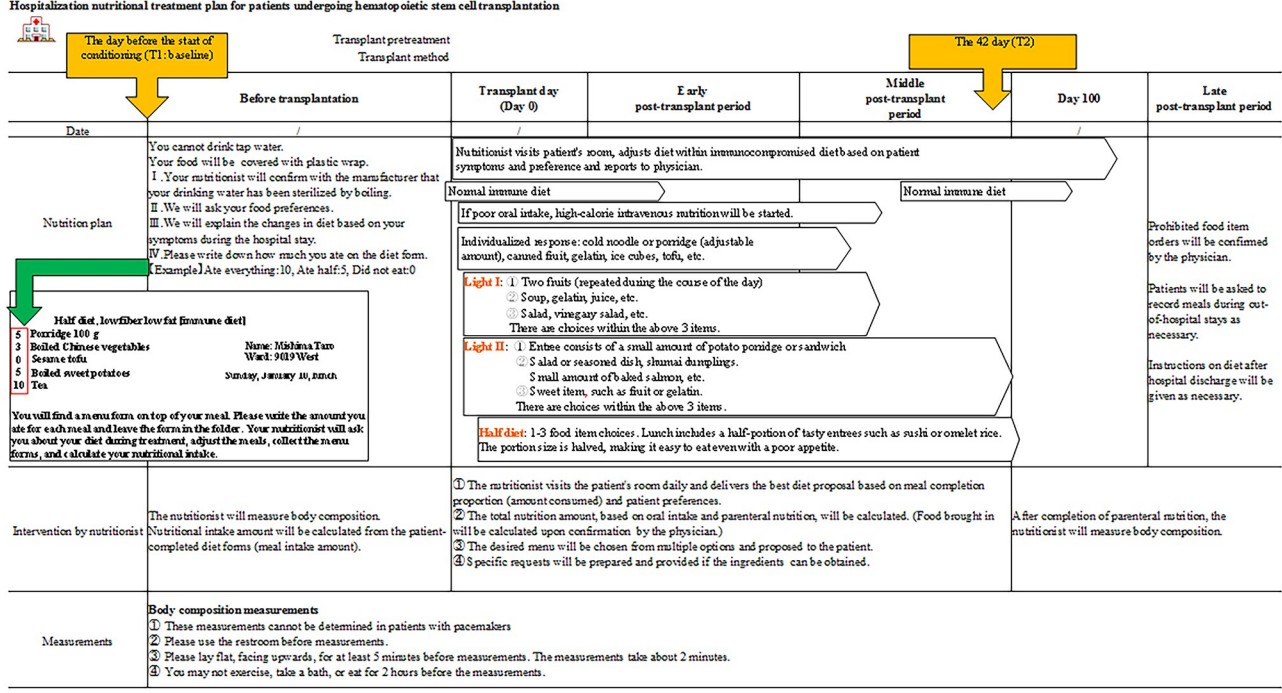

**Fig 1. The nutritional pathway implemented based on nutritional guidance by the Shizuoka Cancer Center blood and hematopoietic cell transplantation team.**

(SMM), fat mass, and phase angle (PA) [21, 22]. Loss of skeletal muscle mass (%LSMM) and loss of fat mass (%LFM) were calculated, and we examined their associations with %LBW. We also calculated the %LSMM of the body extremities and the trunk and evaluated their associations with %LBW.

Similarly, we calculated the daily total caloric energy supply and the amount of protein (PN, oral intake). The basal energy expenditure rate (BEE rate) per ideal body weight (IBW) [23, 24] was obtained by comparing the BEE rate against the total caloric supply during the study period; the total protein supply per IBW was also calculated. The associations of BEE rate and IBW with %LBW were then evaluated.

The participants were categorized as those with and without weight loss during the study period, and the BEE rate cutoff value was determined.

We evaluated grip strength (T.K.K.5401 GRIP-D Digital Grip Dynamometer; Takei Scientific Instruments Co., Ltd., Tokyo, Japan) and pinch strength (Jamar Pinch Gauge Plus+ Digital—50 lb Capacity; USA) and monitored salivary amylase values (Dry Clinical Chemistry Analyzer 34549000 Nipro Co., Japan) [25] at T1, the engraftment day (Immune power is the lowest with neutrophil counts >500/cm$^3$), and T2.

To assess the patient burden during the study period, we evaluated the temporal relationships of nutrition-related adverse events (emesis, nausea, decreased appetite, mucosal damage, and taste disorder; Common Terminology Criteria for Adverse Events [CTCAE] v.5.0) [26], the sum of each grade (e.g., Grade3 = 3) for each day of the treatment course divided by the case number (Table 1), total oral caloric intake/IBW (divided by case number for each day of treatment), and PS (for each day of the treatment, e.g., Grade 2 = 2) divided by the number of

**Table 1. Nutrition-related adverse events according to the CTCAE and PS.**

| CTCAE v5.0 Term | Grade 1 | Grade 2 | Grade 3 | Grade 4 | Grade 5 | CTCAE v5.0 AE Term Definition |
|---|---|---|---|---|---|---|
| Mucositis oral | Asymptomatic or mild symptoms; intervention not indicated | Moderate pain or ulcer that does not interfere with oral intake; modified diet indicated | Severe pain; interfering with oral intake | Life-threatening consequences; urgent intervention indicated | Death | A disorder characterized by ulceration or inflammation of the oral mucosal. |
| Nausea | Loss of appetite without alteration in eating habits | Oral intake decreased without significant weight loss, dehydration or malnutrition | Inadequate oral caloric or fluid intake; tube feeding, total parenteral nutrition or hospitalization indicated | - | - | A disorder characterized by a queasy sensation and/or the urge to vomit. |
| Vomiting | Intervention not indicated | Outpatient IV hydration; medical intervention indicated | Tube feeding, total parenteral nutrition, or hospitalization indicated | Life-threatening consequences | Death | A disorder characterized by the reflexive act of ejecting the contents of the stomach through the mouth. |
| Anorexia | Loss of appetite without alteration in eating habits | Oral intake altered without significant weight loss or malnutrition; oral nutritional supplements indicated | Associated with significant weight loss or malnutrition (e.g., inadequate oral caloric and/or fluid intake); tube feeding or total parenteral nutrition indicated | Life-threatening consequences; urgent intervention indicated | Death | A disorder characterized by a loss of appetite. |
| Dysgeusia | Altered taste but no change in diet | Altered taste with change in diet (e.g., oral supplements); noxious or unpleasant taste; loss of taste | - | - | - | A disorder characterized by abnormal sensual experience with the taste of foodstuffs; it may be related to a decrease in the sense of smell. |
| Oropharyngeal pain | Mild pain | Moderate pain; altered oral intake; non-narcotics initiated; topical analgesics initiated | Severe pain; severely altered eating/swallowing; narcotics initiated; requires parenteral nutrition | - | - | A disorder characterized by a sensation of marked discomfort in the oropharynx. |

Table 2. **Eastern Cooperative Oncology Group (ECOG) performance status.**

| Grade ECOG | |
|---|---|
| 0 | Fully active, able to carry on all pre-disease performance without restriction |
| 1 | Restricted in physically strenuous activity but ambulatory and able to carry out work of a light or sedentary nature, e.g., light house work, office work |
| 2 | Ambulatory and capable of all selfcare but unable to carry out any work activities. Up and about more than 50% of waking hours |
| 3 | Capable of only limited selfcare, confined to bed or chair more than 50% of waking hours |
| 4 | Completely disabled. Cannot carry on any selfcare. Totally confined to bed or chair |
| 5 | Dead |

patients for that day (Tables 1 and 2). Associations between daily oral caloric intake and the BEE rate were assessed on the start date of oral intake and at T2.

Regarding serum albumin (Alb) and C-reactive protein (CRP) before the start of conditioning (i.e., within two weeks before T1), we assessed the maximum laboratory values, as well as the number of days these values were maintained. We also evaluated the associations between these two parameters and investigated the associations between changes in Alb and CRP level with %LBW.

We investigated the mortality rate at two years (following day 0) and the survival time from the start of treatment. For surviving cases at two years, we examined the SMM [27] at T1 via BIA, the PA [28], the hematopoietic cell transplant-comorbidity index (HCT-CI) [29], and graft-versus-host diseases (GVHDs) during the study period [30].

Measurements via the InBody S20® device were taken 2 h after breakfast (from 10:00 a.m. to 12:00 p.m.). We set the reference extracellular fluid/total body fluid ratio as 0.35 and the extracellular water/total body water ratio as 0.40. We defined the upper limits of the extracellular fluid/total body fluid ratio and extracellular water/body water ratio (indicating mild edema) as 0.35–0.38 and 0.40–0.43, respectively. If edema was noted, the measurements were re-performed the next day as edema may influence measurements. All variables were measured using the high-precision body composition analyzer, InBody; the rate for each variable was calculated [21, 22].

The amount of oral nutritional intake was calculated based on information retrieved from meal card records (i.e., intake rates evaluated in proportions; Fig 1). We then collected data from electronic medical records (organized into an Excel chart), along with the amount of PN.

The salivary amylase monitor showed diurnal variations (for amylase activity, the amount of enzyme that produces 1 μmol of maltose-equivalent reducing sugar per min at 37˚C is shown as 1 unit). According to the device specifications, we confirmed that the body temperature was below 37˚C 1–2 h after lunch (at 2:00 p.m.). A saliva collection paper was then inserted into the oral cavity, and saliva was collected twice directly from the sublingual region for 30 s. After monitoring the salivary amylase activity, the grip strength and pinch strength of the dominant arm were alternately measured in the sitting position three times each at 1-min intervals. The pinch strength was determined as the thumb to index finger pulp pinch with the elbow flexed at 90 degrees.

## Statistical analysis

For all evaluation items, the median, minimum, and maximum values were used to evaluate normality (Shapiro–Wilk test) [31]. The receiver operating characteristic curve was used to

evaluate the BEE rate cutoff value (obtained from the presence or absence of weight loss). The %LSMM of the limbs and trunk, grip strength and pinch strength, and salivary amylase activity values (kU/L) were evaluated using analysis of variance (ANOVA: post hoc test). Pearson's product-moment correlation was used to evaluate the association between body composition and nutritional supply, nutrition-related adverse events over time, and associations between oral caloric intake and PS. To evaluate treatment-related toxicity, pretreatment Alb and CRP were compared with change values during the treatment course using the Wilcoxon signed-rank test. Pearson's product-moment correlation was used for the association between both test values and %LBW. Logistic multivariate analysis was used to examine two-year survival and non-survival SMM, PA, HCT-CI, and GVHD. We used JMP version 12.0 for Windows (SAS Institute, Cary, NC, USA) for statistical analyses, and the level of statistical significance was set to $p < 0.05$.

## Ethical considerations

This paper used data from prospective clinical studies approved by the Shizuoka Cancer Center Institutional Review Board (SCC IRB), combined with retrospectively obtained outcome data; this secondary study was approved by the SCC IRB (approval number: T26-57). The patients were educated using an explanatory document developed for this study and were registered only after obtaining their written informed consent. Consent to collect data on outcomes was obtained using bulletin boards at the hospital (https://www.scchr.jp/clinicaltrial/index.html). Though not available in a public repository, the data used for this study will be made available to other researchers upon reasonable request.

## Results

From June 2015 to June 2019, 92 patients underwent HSCT and were eligible for the evaluated nutritional pathway intervention during the study period. The flowchart for patient selection and enrollment process is shown in Fig 2. Table 3 shows the demographic and medical data of the 40 patients included in the current study. Table 4 shows the results for the clinical nutrition indicators examined during the evaluation period.

For participants enrolled in the nutritional pathway, the mean BMI at T1 was 21.6 kg/m$^2$ (range: 19.0–24.7), and the mean %IBW was 98.1% (range: 84.2–112.3). We found a correlation between %LBW and %LSMM ($r = 0.70$, $p < 0.001$; Fig 3), but no correlation was observed between %LBW and %LFM ($r = 0.20$, $p = 0.21$). The %LSMM in the limbs and trunk were equal ($p = 0.91$, ANOVA), and the %LSMM was correlated with %LBW in all locations (right arm, $r = 0.78$, $p < 0.01$; left arm, $r = 0.65$, $p < 0.01$; trunk, $r = 0.77$, $p < 0.01$; right leg, $r = 0.40$, $p < 0.01$; and left leg, $r = 0.39$, $p < 0.01$).

We also found a correlation between %LBW and the BEE rate ($r = 0.49$, $p < 0.01$; Fig 3) but no evidence of an association with the amount of protein supplied ($r = 0.26$, $p = 0.10$). The BEE rate cutoff value between the group with weight loss (n = 27) and the group without weight loss (n = 13) was 1.1 based on the receiver operating characteristic curve (Fig 4).

The variances for grip strength, pinch strength, and salivary amylase monitor measurements (KU/L) at T1, at each patient's engraftment day (mean: day 18, range: day 14–40), and at T2 were similar (Fig 5).

Nutrition-related adverse events during the study period were correlated with oral caloric intake and PS ($r = -0.93$, $p < 0.01$ and $r = 0.91$, $p < 0.01$, respectively; Fig 6). We observed an association between the start date of oral intake and the BEE rate for daily oral caloric intake at T2 ($r = 0.32$, $p = 0.03$).

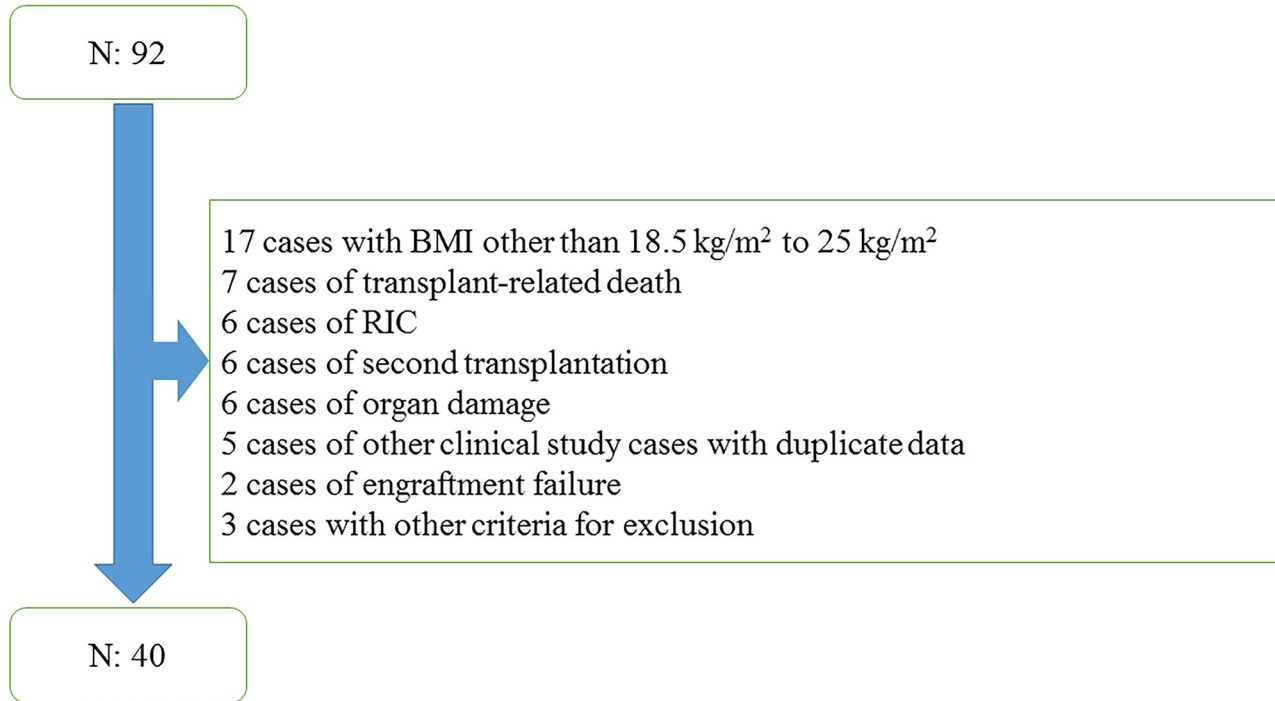

**Fig 2. Flowchart of the patient selection and enrollment process.**

Alb and CRP levels before treatment initiation (i.e., before the start of conditioning) changed significantly before and after engraftment, and the changes in both test values were negatively correlated ($r = -0.40$, $p = 0.01$; Fig 7).

Changes in Alb ($-1.1$ g/dL, range: $-2.2$–$0$) and changes in CRP (7.27 mg/dL, range: 0.09–22.14) were not correlated with changes in %LBW ($-3.9$, range: $-11.0$–$6.3$) ($r = 0.27$, $p = 0.09$ and $r = 0.13$, $p = 0.42$, respectively).

**Table 3. Patient background.**

| Study period | June 2015–June 2019 |
|---|---|
| Sample size, n (male/female) | 40 (20/20) |
| Age (min-max) | 52 (20–68) |
| Disease | |
| Acute myeloid leukemia | 13 |
| Myelodysplastic syndrome | 12 |
| Malignant lymphoma | 8 |
| Acute lymphoblastic leukemia | 6 |
| Chronic myeloid leukemia | 1 |
| Conditioning regimen | |
| Busulfan ($>6.4$ mg/kg) | 25 |
| Total body irradiation ($\geq$5Gy in a single fraction, $\geq$8Gy in multiple fractions) | 14 |
| Melphalan ($>140$ mg/m$^2$) | 1 |
| Transplant source | |
| Unrelated bone marrow transplantation | 25 |
| Cord blood transplantation | 8 |
| Peripheral blood stem cell transplantation | 7 |
| Non-inherited maternal antigen | 20 |

**Table 4. Clinical indicators assessed during the nutritional pathway evaluation.**

| Clinical indicator | value | *p |
|---|---|---|
| Preoperative BMI (range) | 21.6kg/m$^2$ (19.0–24.7) | 0.40 |
| %LBW (range): T1→T2 | −3.9 (−11.0–6.3) | 0.36 |
| SMM (range): T1 | 22.8kg (16.9–31.9) | <0.01 |
| %LSMM (range): T1→T2 | −2.9 (−12.1–14.6) | 0.38 |
| Right upper limb: %LSMM (range) | −2.8 (−18.0–18.1) | 0.12 |
| Left upper limb: %LSMM (range) | −4.1 (−18.1–47.7) | <0.01 |
| Trunk: %LSMM (range) | −2.1 (−11.4–14.9) | 0.32 |
| Right lower limb: %LSMM (range) | −3.4 (−20.7–76.2) | <0.01 |
| Left lower limb: %LSMM (range) | −2.8 (−18.5–38.0) | <0.01 |
| FM (range): T1 | 15.0kg (4.0–21.0) | 0.28 |
| %LFM (range): T1→T2 | −8.8 (−35.7–42.9) | <0.05 |
| BIA: ECF/TBF (range): T1 | 0.35 (0.33–0.37) | 0.27 |
| BIA: ECF/TBF (range): T2 | 0.40 (0.38–0.42) | <0.01 |
| BIA: ECW/TBW (range): T1 | 0.35 (0.29–0.38) | 0.62 |
| BIA: ECW/TBW (range):T2 | 0.40 (0.33–0.42) | <0.01 |
| PA (range): T1→T2 | 4.83 (3.31–6.27) | 0.96 |
| PA (range): T1→T2 | −0.51 (−1.57–0.53) | 0.52 |
| Total calorie intake (range): T1→T2 | 24kcal/IBW/day (17–30) | 0.53 |
| PN calorie intake (range): T1→T2 | 13kcal/IBW/day (2–24) | 0.82 |
| Orally ingested calories (range): T1→T2 | 11kcal/IBW/day (3–22) | 0.15 |
| Total protein intake (range): T1→T2 | 0.8g/IBW/day (0.5–1.2) | 0.19 |
| PN protein intake (range): T1→T2 | 0.5g/IBW/day (0.1–0.8) | 0.08 |
| Orally ingested protein intake (range): T1→T2 | 0.3g/IBW/day (0.1–0.9) | <0.01 |
| Oral intake initiation day (range) | day 16 (0–38) | <0.01 |
| PN energy rate (range): T1→T2 | 56% (10–89) | 0.39 |
| PN off elapsed day | day 41(21–84) | <0.01 |
| BEE (range) | 1,330 kcal/IBW/day (1,097–1,684) | <0.05 |
| BEE rate/IBW % (range): T1→T2 | 100% (72–139) | 0.93 |
| Daily oral energy intake BEE sufficiency rate at T2 (day 42) (range) | 84% (6–162) | 0.43 |
| Engraftment (range) | day 18 (14–40) | <0.01 |

BMI, body mass index; LBW, loss of body weight; SMM skeletal muscle mass; %LSMM, loss skeletal muscle mass; FM fat mass; %LFM, percent loss of fat mass; BIA, bioelectrical impedance analysis; ECF, extracellular fluid; TBF, total body fluid; ECW, extracellular water; TBW, total body water; PA, Phase angle; IBW, ideal body weight; PN, parenteral nutrition; BEE, basal energy expenditure; the energy intake sufficiency rate: [(PN calories, orally ingested calories, or enhanced nutrition calories) / total energy intake] × 100.

*p: Shapiro–Wilk

There were 10 cases of mortality (relapse of the malignancy in nine cases, and graft-versus-host disease in one case) at the two-year follow-up (starting from day 0), and the mean survival time starting from day 0 was 230 days (range: 99–523). We found an association between SMM at T1 and two-year survival (30 surviving cases; odds ratio: 1.2, 95% confidence interval: 1.014–1.495, $p$ = 0.04). However, SMM at T1 was not associated with PA, HCT-Cl, or the occurrence of GVHDs during the study period (Fig 8). Edema was not observed to affect SMM during examinations via the BIA method at T1 (Table 3).

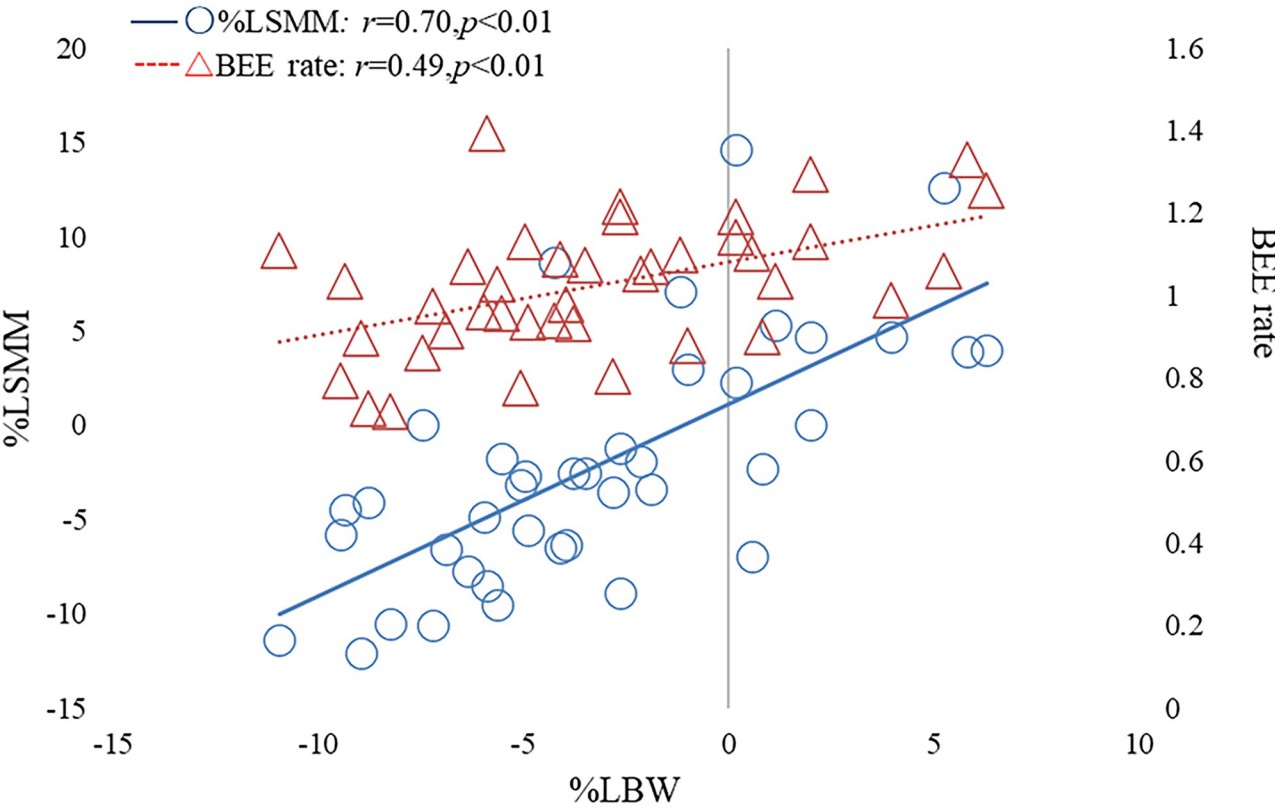

**Fig 3. Weight loss rate: %LBW, skeletal muscle loss rate: %LSMM, and basal metabolic calorie sufficiency rate: BEE rate from T1 to T2.**

## Discussion

In this paper, we examined the validity of nutritional pathway treatment by examining the patient's clinical burden and adverse outcomes and explored the associations between clinical outcomes and nutritional indicators.

We found a correlation between %LBW, %LSMM, and the BEE rate over time, but there was no association with the amount of protein supplied. Examining the BEE rate in groups of patients with or without weight loss demonstrated a cutoff value of 1.1. Our results showed that the observed correlations between %LBW and %LSMM were supported by the observed BEE rate of 1.1. This was similar to the results from a previous report that showed a statistically significant positive correlation between caloric intake and nitrogen balance [32]. In the field of nutrition, the protein-saving effect of caloric intake is well known [33]; nitrogen balance improves as caloric intake increases, although calorie deficiency reduces the efficiency of protein utilization [34]. Based on this general information, the amount of nutrition supply within HSCT should be considered when planning treatment. Since the %LBW values of the limbs and trunk were correlated in the current study, it is likely that a decrease in muscle mass occurred throughout the body and that the decrease in PS described later in this manuscript may have caused a decrease in muscle mass. Therefore, the logic is to improve muscle mass loss by increasing PN (energy) and increasing BEE rate to suppress weight loss. However, hyperglycemia is frequently observed when PN is prolonged, and studies suggest that prolonged PN is associated with the onset of infectious diseases [35, 36]. It has also been reported that excessive fluid administration via PN adversely affects engraftment syndrome (which is

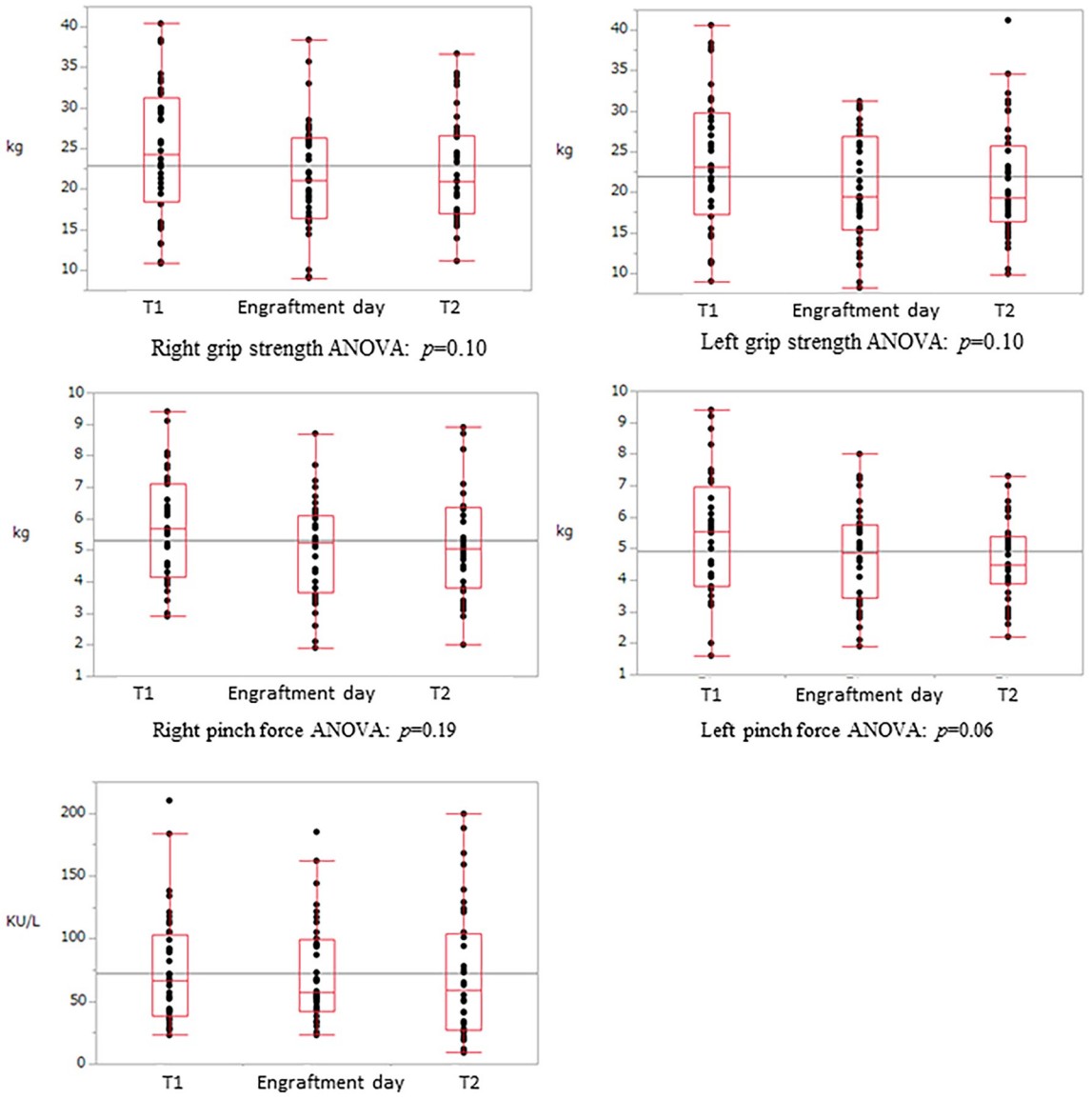

| | T1 | Engraftment day | T2 |
|---|---|---|---|
| Right pinch force (range) | 4.2kg (2.9-8.1), $p$=0.33 | 4.4kg (1.9-5.8), $p$=0.51 | 3.9kg (2.9-5.9), $p$=0.08 |
| Left pinch force (range) | 3.8kg (1.6-6.6), $p$=0.59 | 3.6kg (1.9-5.8), $p$=0.80 | 4.0kg (2.6-5.4), $p$=0.62 |
| Right grip strength (range) | 19.1kg (10.8-29.3), $p$=0.33 | 17.2kg (9.0-25.6), $p$=0.51 | 17.2kg (11.1-26.4), $p$=0.08 |
| Left grip strength (range) | 17.6kg (9.0-28.0), $p$=0.65 | 16.3kg (8.2-26.1), $p$=0.16 | 17.2kg (9.9-25.1), $p$=0.08 |
| Saliva amylase monitor measured value (range) | 57KU/L (23-210), $p$<0.01 | 53KU/L (23-162), $p$<0.01 | 41KU/L (12-188), $p$<0.01 |

$p$: Shapiro-Wilk

**Fig 4. Grip and pinch strength and salivary amylase activity values at T1, on the engraftment date, and at T2.** Lower right photo: Left, grip strength meter; upper right, salivary amylase monitor; lower right, pinch strength meter.

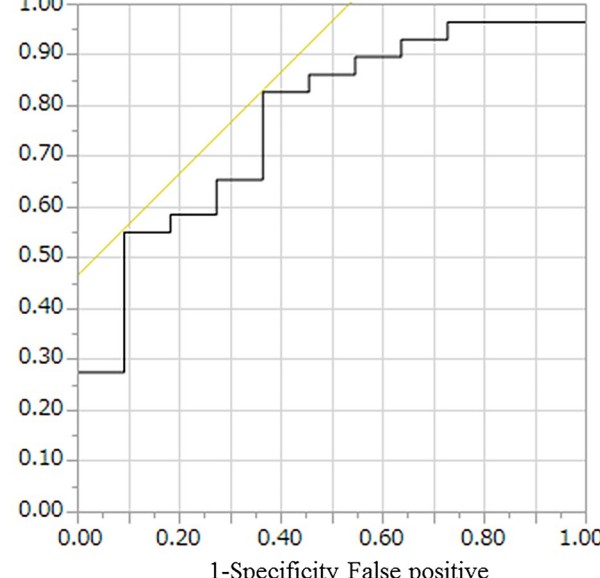

|  | BEE rate ≥1.1 | BEE rate <1.1 |
|---|---|---|
| LBW | 82% | 18% |
| None LBW | 36% | 64% |

Area under the curve: AUC 0.77116
Sensitivity: 0.8276, 1-Specificity: 0.3636
Cut off of BEE rate: 1.1

**Fig 5. Receiver operating characteristic curve for weight loss and basal energy expenditure sufficiency rate from T1 to T2.**

observed in HSCT) [37]. Therefore, a rapid transition from PN to oral intake is desirable. Nutritional interventions that vary according to the course of treatment and adhere to the nutritional pathway evaluated in this paper are likely beneficial.

Using a salivary amylase monitor [25], we examined whether the results supported the use of ST, a stress coefficient associated with the BEE rate in MAC. We found that the variances of the mean values of salivary amylase before treatment, on the engraftment day, and on day 42 were equal. This suggests that a nutritional metabolism BEE rate of 1.1 (with evidence under certain stress) is supported as a stress coefficient.

| Factor | | Odds rate | 95%CI | | p |
|---|---|---|---|---|---|
| SMM: T1 | | 1.2 | 1.014 | 1.495 | 0.03 |
| Pa: T1 | | 0.8 | 0.237 | 2.862 | 0.77 |
| HCT-CI (Score): T1 | 0-1 | 0.4 | 0.054 | 3.634 | 0.39 |
| | 0-2 | 0.2 | 0.128 | 23.465 | 0.94 |
| | 0-3 | 0.3 | 0.01 | 7.571 | 0.39 |
| GVHD (Grade): T1→T2 | 0-1 | 2.0 | 0.352 | 16.071 | 0.45 |
| | 0->2 | 2.5 | 0.455 | 19.738 | 0.30 |

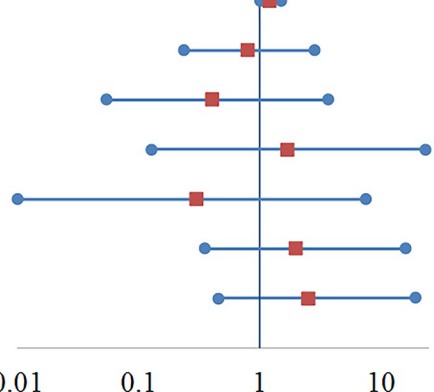

SMM: skeletal muscle mass
Pa: Phase-angle
HCT-CI: hematopoietic cell transplant-comorbidity
GVHD: graft-versus-host disease

**Fig 6. Association between two-year survival, mortality, and clinical indicators.**

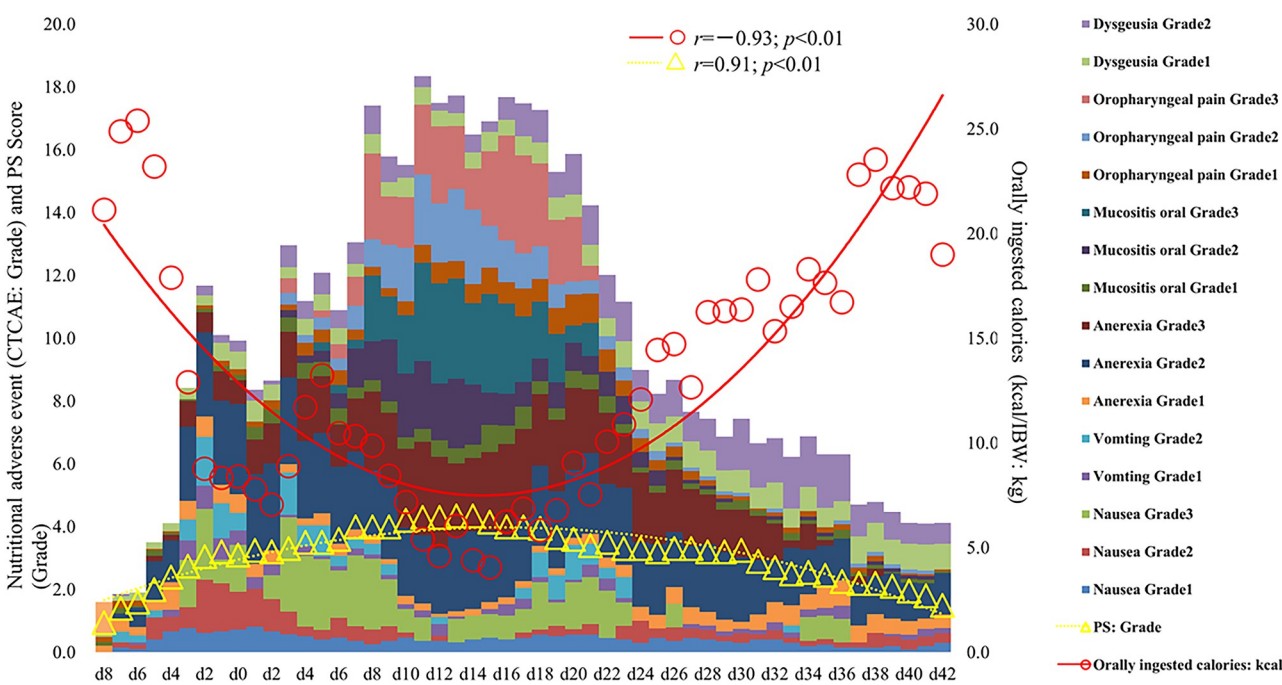

**Fig 7. Nutrition-related adverse events and oral caloric intake from T1 to T2 and standard weight and performance status assessments over time.**

In the nutritional evaluation within this study, %LSMM was associated with %LBW in transplant patients over an approximately two-year period and was also correlated with % LSMM in the limbs and trunk; however, there was no change in grip strength and pinch strength over time (Fig 4). The relationship between %LSMM in HSCT and muscle strength remains unclear based on the results presented in this paper [38].

Since we found a positive correlation between nutrition-related adverse events and PS over time, it is likely that in the current study, the total energy expenditure declined due to a decrease in PS [39, 40]. In addition, the amount of oral intake decreased and showed a negative correlation due to the overlap of nutrition-related adverse events and oral caloric intake from T1 to T2 (Fig 6), suggesting that diet-induced energy production decreased during the treatment course [41]. We observed that oral caloric intake improved during the study period, and we observed a reduction in nutrition-related adverse events. This result is supported by the observed correlation between the oral intake start date and the oral intake BEE rate at T2 (day 42). This suggests the importance of consistent follow-up for various symptoms during treatment, starting from early nutrition intervention within the nutritional pathway treatment.

The diet before the treatment initiation for patients undergoing HSCT was a normal diet (i.e., a solid food diet). The nutritional pathway, due to the overlapping nutrition-related adverse events shown in Fig 7, consistently included jelly that does not diffuse like water to the upper digestive tract, as well as sorbet and ice cubes with an oral phase for mitigating nausea and loss of appetite, which are the chief complaints and dietary needs of patients at the time of appetite recovery (i.e., when they start eating orally during the treatment course, starting from the time of nutritional guidance at T1; Fig 1). The therapeutic value and goals of the nutritional pathway are to reduce nutrition-related adverse events, to take into consideration the proportion of intake, as well as the patient preferences described on the meal card, and to sequentially adjust the patient to normal meals to obtain greater caloric intake (with the consent of the

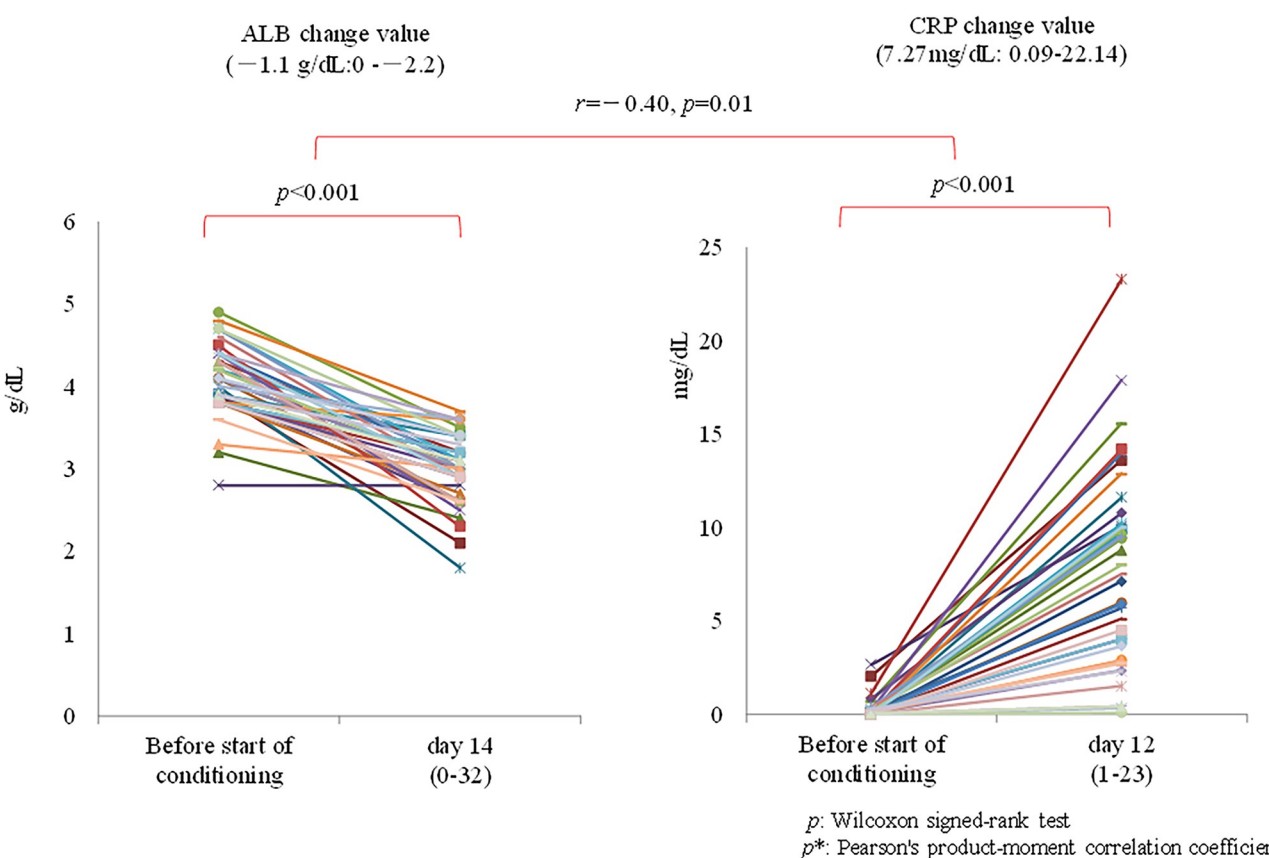

**Fig 8. Changes in and associations with serum albumin and C-reactive protein values.**

patient) [15]. Our results agree with the evidence from time-course nutritional interventions and monitoring tailored to the patient's burden. All findings suggest that a decrease in PS is associated with an increase in nutrition-related adverse events. In addition, a previous systematic review presents a discussion of nutritional counseling and its outcomes among patients undergoing cancer chemotherapy (other than for the treatment of hematopoietic tumors) with a focus on weight loss; the consistency between the results of this systematic review and the monitoring and intervention presented in this paper is objectively demonstrated by the current results [42].

During the treatment course, Alb and CRP levels before and after engraftment changed significantly; the changes in both test values were negatively correlated, although an improvement in both values was observed. We found no evidence of an association between changes in Alb or CRP and changes in %LBW. This result is likely due to the induction of inflammatory cytokines in engraftment syndrome [43] and a biological reaction (cytokine syndrome) similar to that following surgical invasion [44]. However, in this study, there was no association between the %LBW, total caloric supply, and %LSMM. In other words, changes in Alb and CRP were observed (as in the case of the unavoidable increase in protein catabolism and the suppression of protein synthesis during the perioperative period) [45]; however, this is unlikely to be a factor directly related to %LBW based on our results.

Among the clinical nutritional indicators for two-year mortality and survival (starting from day 0), SMM at T1 was noted as an independent risk factor. Thus far, the outcomes of LBW

[46], the third lumbar skeletal muscle area (measured via the BIA method) [27], PA [28], and sarcopenia [47] have been discussed. However, to the best of our knowledge, this paper is the first to suggest that SMM may affect survival [48]. Hence, it is necessary to monitor SMM and follow up with a nutritional management plan [49] starting at the time of induction therapy leading up to HSCT and consolidation therapy.

HSCT was performed for the first time by Thomas et al. in 1957 [50], and various treatments have since been developed [1]. However, the five-year survival rate for hematopoietic cancer is 60%, which is low compared to many other cancers [51]. In this paper, we clarified that the nutritional domain [49, 52, 53] is involved in HSCT and examined the significance of intervention via a nutritional pathway. As a result, in MAC for HSCT (aiming at remission and using a fixed treatment period [day 42] based on past results, i.e., the PN period) [7, 8, 15, 17–19], we clarified that a BEE rate of 1.1 is required for body weight maintenance based on salivary amylase values that were consistent with the observed correlations between body weight, skeletal muscle mass, and the amount of caloric supply. In addition, we observed a correlation between nutrition-related adverse events over time and oral caloric intake, and an early oral intake start date was related to the BEE rate for oral caloric intake at T2 (day 42). This clearly shows that the improvement in the proportion of food intake through a consistent intervention based on past results leads to an improved treatment course after transplantation and pain relief for the patient. In other words, these results suggest that the nutritional pathway examined in this study is valid and may be useful for the formation of an efficient patient-oriented nutritional intervention platform for HSCT requiring long-term PN.

Our results support the validity of a secured nutritional pathway with a BEE rate of 1.1 for HSCT pretreatment with MAC. Specifically, we demonstrated that this pathway prevents weight loss in response to nutrition-related adverse events and that SMM before treatment is an independent risk factor for reduced 2-year survival.

A limitation of this study is that the effects of immunosuppressants and steroids on GVHD have not previously been investigated within randomized controlled trials. These outcomes require further investigations by studies that enroll more cases to examine 5-year survival rates.

## Conclusions

Our results support the conclusion that treatment via a nutritional pathway in MAC as a pretreatment for HSCT is valid and effective and that SMM before treatment initiation is an independent factor predicting 2-year survival outcomes.

## Acknowledgments

We would like to thank Editage (www.editage.com) for English language editing.

The author would like to thank all registered dietitian at the Shizuoka Cancer Center, Japan, for assistance with data collection in nutritional pathway.

## Author Contributions

**Conceptualization:** Takashi Aoyama.

**Data curation:** Takashi Aoyama.

**Formal analysis:** Takashi Aoyama, Akifumi Notsu.

**Funding acquisition:** Takashi Aoyama.

**Investigation:** Takashi Aoyama, Takashi Yurikusa, Koki Ichimaru, Masanori Tsuji, Kanako Yoshitsugu, Masafumi Fukaya, Terukazu Enami, Takashi Ikeda.

**Methodology:** Takashi Aoyama.

**Project administration:** Takashi Aoyama.

**Resources:** Takashi Aoyama.

**Software:** Takashi Aoyama.

**Supervision:** Takashi Aoyama, Osamu Imataki, Takashi Ikeda.

**Validation:** Takashi Aoyama.

**Visualization:** Takashi Aoyama.

**Writing – original draft:** Takashi Aoyama.

**Writing – review & editing:** Takashi Aoyama.

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
