## [Decision Letter · Decision Letter 0]

6 Dec 2021

PONE-D-21-25328Examination of a nutritional treatment pathway according to pretreatment health status and stress levels of patients undergoing hematopoietic stem cell transplantationPLOS ONE

Dear Dr. Aoyama,

Thank you for submitting your manuscript to PLOS ONE. After careful consideration, we feel that it has merit but does not fully meet PLOS ONE’s publication criteria as it currently stands. Therefore, we invite you to submit a revised version of the manuscript that addresses the points raised during the review process.

We have received the opinions of expert reviewer and we invite you to submit a revised version of the manuscript, please consider and address each of the comments raised by the reviewers.  

We look forward to receiving your revised manuscript.

Kind regards,

Senthilnathan Palaniyandi, Ph.D

Academic Editor

PLOS ONE

Journal Requirements:

No

No

Reviewers' comments:

Reviewer's Responses to Questions

**Comments to the Author**

1. Is the manuscript technically sound, and do the data support the conclusions?

Reviewer #1: Yes

Reviewer #2: Yes

2. Has the statistical analysis been performed appropriately and rigorously? 

Reviewer #1: I Don't Know

Reviewer #2: Yes

3. Have the authors made all data underlying the findings in their manuscript fully available?

Reviewer #1: Yes

Reviewer #2: Yes

4. Is the manuscript presented in an intelligible fashion and written in standard English?

Reviewer #1: Yes

Reviewer #2: Yes

5. Review Comments to the Author

Reviewer #1: The authors' study underlines the importance of nutrition in allogeneic transplantation. This study is from Shizuoka in Japan. The patient numbers are low (40 patients treated with a myeloablative transplant). The authors claim that skeletal mass before transplant is an independent risk factor for death at 2 years, however they do not indicate what were the causes of death. This is important and should go into a multivariate analysis. Common causes of death after transplant are: infection, graft versus host disease, relapse of the malignancy. The literature is not up to-date. For example they fail to mention: Eglseer D et al recently published in BMT (341 patients), Tamaki M et al recently published in TCT (182 patients). The English needs improvements, for example in the legend of Fig 2 they indicate "3 cases of other cases" (??)

Reviewer #2: The abstract section should be restructured into Introduction, Methods, Results, and Conclusion.

Other parts of the manuscript is written well and I don't finy and major issues with the study pther than a relatively low sample size of 40 patients. The results of this studya re also important clinically and provide some advancement in the field of hematopoietic stem cell transplantation.

6. PLOS authors have the option to publish the peer review history of their article (what does this mean?). If published, this will include your full peer review and any attached files.

Reviewer #1: No

Reviewer #2: No

---

## [Author Response · Author response to Decision Letter 0]

3 May 2022

Reviewer comments

Reviewer #1: The authors' study underlines the importance of nutrition in allogeneic transplantation. This study is from Shizuoka in Japan. The patient numbers are low (40 patients treated with a myeloablative transplant). The authors claim that skeletal mass before transplant is an independent risk factor for death at 2 years, however they do not indicate what were the causes of death. This is important and should go into a multivariate analysis. Common causes of death after transplant are: infection, graft versus host disease, relapse of the malignancy. The literature is not up to-date. For example they fail to mention: Eglseer D et al recently published in BMT (341 patients), Tamaki M et al recently published in TCT (182 patients). The English needs improvements, for example in the legend of Fig 2 they indicate "3 cases of other cases" (??)

Reply: Thank you for your comment. We have revised the Results and Discussion sections and Figure 2 in order to include more background information.

Reviewer #2: The abstract section should be restructured into Introduction, Methods, Results, and Conclusion.

Reply: Thank you for your comment. We have revised the abstract to include this structured format.

---

## [Decision Letter · Decision Letter 1]

7 Jul 2022

Examination of a nutritional treatment pathway according to pretreatment health status and stress levels of patients undergoing hematopoietic stem cell transplantation

PONE-D-21-25328R1

Dear Dr. Aoyama,

We’re pleased to inform you that your manuscript has been judged scientifically suitable for publication and will be formally accepted for publication once it meets all outstanding technical requirements.

Kind regards,

Senthilnathan Palaniyandi, Ph.D

Academic Editor

PLOS ONE

Additional Editor Comments (optional):

Reviewers' comments:

Reviewer's Responses to Questions

**Comments to the Author**

1. If the authors have adequately addressed your comments raised in a previous round of review and you feel that this manuscript is now acceptable for publication, you may indicate that here to bypass the “Comments to the Author” section, enter your conflict of interest statement in the “Confidential to Editor” section, and submit your "Accept" recommendation.

Reviewer #1: All comments have been addressed

Reviewer #2: All comments have been addressed

2. Is the manuscript technically sound, and do the data support the conclusions?

Reviewer #1: Yes

Reviewer #2: Yes

3. Has the statistical analysis been performed appropriately and rigorously? 

Reviewer #1: Yes

Reviewer #2: Yes

4. Have the authors made all data underlying the findings in their manuscript fully available?

Reviewer #1: Yes

Reviewer #2: Yes

5. Is the manuscript presented in an intelligible fashion and written in standard English?

Reviewer #1: Yes

Reviewer #2: Yes

6. Review Comments to the Author

Reviewer #1: Manuscript is improved, therefore acceptable for publication, nutrition is an important topic, patient numbers still low, but sufficient for a pilot study

Reviewer #2: The authors have addressed my prior comments and other reviewer's comments. I believe the manuscript is suitable for publication at its current state.

7. PLOS authors have the option to publish the peer review history of their article (what does this mean?). If published, this will include your full peer review and any attached files.

Reviewer #1: No

Reviewer #2: No

---

## [Editor Report · Acceptance letter]

22 Jul 2022

PONE-D-21-25328R1 

Examination of a nutritional treatment pathway according to pretreatment health status and stress levels of patients undergoing hematopoietic stem cell transplantation 

Dear Dr. Aoyama:

I'm pleased to inform you that your manuscript has been deemed suitable for publication in PLOS ONE. Congratulations! Your manuscript is now with our production department. 

Kind regards, 

on behalf of

Dr. Senthilnathan Palaniyandi 

Academic Editor

PLOS ONE